# UNIFIED UNCERTAIN DUAL-PROMPTS CROSS-DOMAIN SEGMENTATION FRAMEWORK FOR MEDICAL IMAGE SEGMENTATION

## ABSTRACT

Unsupervised cross-domain segmentation addresses the challenge of label dependence in cross-domain medical image segmentation. Yet, most existing methods treat domain adaptation and segmentation as ***Two Separate Steps*** and primarily focus on global domain adaptation, lacking the ability to prioritize segmentation-specific information during domain adaptation. Additionally, for cross-domain segmentation, extracting domain-invariant feature representation remains an unavoidable challenge. These challenges significantly reduce segmentation performance. To this end, we propose a novel Unified Uncertain Dual-prompts cross-domain Segmentation framework (UUDS) for unsupervised cross-domain medical image segmentation. Specifically, our UUDS forms a unified framework by integrating domain adaptation and segmentation models, facilitating interaction between the two tasks, addressing the challenge of emphasizing segmentation semantics while domain adaptation. Additionally, UUDS creatively uses dual-prompts, domain and segmentation prompts, to ensure that the model can learn domain-invariant feature representations from the cross-domain space. Furthermore, to further facilitate interaction between the two tasks, UUDS uses uncertainty estimation to dynamically compute segmentation labels, enabling direct supervision of the cross-domain adaptation process. Extensive experimental results on two representative unsupervised cross-modality medical image segmentation demonstrate that UUDS outperforms state-of-the-art methods, highlighting its effectiveness in addressing domain shifts and marking a significant breakthrough in domain adaptation.

## 1 INTRODUCTION

Unsupervised cross-domain segmentation (Wu et al., 2024) is a promising approach to tackling the challenge of domain shift in cross-modality medical image segmentation. Given the high costs associated with pixel-level data collection and labeling from medical practitioners, it can reduce reliance on manual labels. Existing unsupervised cross-domain segmentation methods (Zou et al., 2018; Lee et al., 2024) attempt to overcome the domain shift between source and target data by aligning the distribution of source and target data through unsupervised domain adaptation (UDA). Despite the impressive performance achieved in various tasks (Li et al., 2019; Chen et al., 2019), these methods treat domain adaptation and segmentation as ***Two Separate Steps***, lacking the ability to establish the interaction between the two tasks for directly using segmentation information to guide domain adaptation. Moreover, these methods primarily focus on global domain adaptation without prioritizing segmentation-aware feature representation while domain adaptation. This limitation fails to establish effective feedback between domain adaptation and segmentation and reduces the effectiveness of methods focused on the feature distribution of segmentation regions, rendering the model less sensitive to segmentation-specific features. Furthermore, extracting domain-invariant information from cross-domain feature representation space remains challenging due to the entanglement of content and domain information. This challenge is even more pronounced in the medical field, where images often contain complex tissues and organs. Therefore, there is a critical urgent for a segmentation-sensitive unified unsupervised cross-domain segmentation method for cross-modality medical image segmentation.

Recently, large-scale Vision-Language Models (VLMs)(Yao et al., 2024; Yu et al., 2023), particularly the Contrastive Language-Image Pre-training (CLIP) model(Radford et al., 2021), have shown promising performance in aligning cross-modality embedding spaces (Lai et al., 2023; Singha et al., 2023; He et al., 2023; Jia et al., 2021). One of the most significant advantages of VLMs is they align visual features from image to natural language sentences or phrases, rather than closed-end labels. Encapsulated in natural language expressions, vision features can travel across domains while maintaining the same semantic meanings. This important property makes the VLMs an ideal source for obtaining domain-invariant prompts. Therefore, VLMs provide significant potential in achieving non-adversarial domain adaptation to address the well-knowledge challenges associated with adversarial learning-based UDA, where obtaining a stable and globally optimal GAN remains difficult, especially in maintaining balance between the generator and discriminator (Sankaranarayanan et al., 2018). However, to the best of our knowledge, no efforts have been made to utilize VLMs for unsupervised cross-modality medical image segmentation due to the significant challenges involved. Specifically, CLIP is trained on natural image-text pairs, resulting in a substantial domain gap between natural and medical images. This raises two key challenges: 1) how to transfer the rich knowledge learned from natural image-text pairs to the medical imaging field remains an open question. 2) Medical images contain more complex anatomical structures and simple textual prompts like "a photo of a [CT/MR]" are insufficient to accurately describe the intricate content of medical images. Yet, medical image segmentation requires distinguishing between multiple tissues and organs for precise localization and segmentation. Although VLMs hold great potential in the field of medical image analysis, these challenges have left them largely unexplored for unsupervised cross-domain medical image segmentation.

In general, a straightforward way for transferring knowledge across domains involves utilizing the text representations from VLMs as a foundation for further fine-tuning models (Qin et al., 2022). However, due to the non-continuous nature of language hard prompts, directly tuning randomly initialized embedding vectors may receive more robust performance and promise to converge to a local optimum (Lester et al., 2021). While hard prompt embeddings from large pre-trained VLMs can effectively adapt to global-level domain information(Zhou et al., 2022), they tend to be less sensitive to detailed information(Jia et al., 2022). Therefore, improving the utilization of knowledge from VLMs for cross-modality medical image segmentation remains a critical area for exploration. Additionally, fully harnessing the potential of CLIP is another important avenue worth exploring.

To this end, we propose a novel Unified Uncertain Dual-prompts cross-domain Segmentation framework (UUDS) for unsupervised cross-domain medical image segmentation by leveraging CLIP's capability in aligning cross-modality embedding spaces. Specifically, UUDS creates a unified CLIP based framework where segmentation and domain adaptation are seamlessly combined for establishing a segmentation-aware unsupervised cross-domain medical image segmentation framework. Furthermore, to overcome CLIP's limitation in describing complex organs and tissues of medical images while maximizing its potential, UUDS innovatively uses dual prompts, domain and segmentation prompts, to learn domain and segmentation invariant feature representation. It simplifies the challenges and complexities involved in prompt learning, addresses the difficulty of describing the intricate content of medical images, and reduces the challenge in learning domain-invariant feature representation from cross-domain feature representation space. Furthermore, UUDS introduces uncertainty estimation to dynamically compute the label of segmentation for directly supervising the cross-domain adaptation, ensuring the semantic information from unlabeled target images can directly supervise the process of domain adaptation and making the model sensitive to segmentation. It facilitates interaction between the two tasks and addresses the limitation that existing methods are unable to directly use the segmentation information for guiding domain adaptation. Experimental results from two public cross-modality medical image domain adaptation and segmentation tasks demonstrate that our UUDS outperforms state-of-the-art UDA methods and performs best on cross-modality domain adaptation and segmentation.

Our main contributions include:

- For the first time, a unified framework for unsupervised cross-domain semantic-aware segmentation by creatively integrating domain adaptation and segmentation models is proposed. It constrains domain adaptation within the segmentation semantic space and addresses the defect of insensitivity to segmentation semantics during the adaptation process.

- The largely vision-language model (CLIP), for the first time, is extended to unsupervised cross-domain medical image segmentation, addressing the significant domain gap challenge of transferring VLMs pre-trained on natural images to medical image field.

- Dual-prompts, domain and segmentation prompts, are proposed to learn domain and segmentation invariant representation learning, simplify the challenges and complexities involved in prompt learning, address the difficulty of describing the intricate content of medical images and the challenge of learning domain-invariant feature representations.

- Novel using uncertainty estimation to dynamically compute the segmentation label for directly supervising the cross-domain adaptation, ensuring the semantic information from unlabeled target images can directly supervise the domain adaptation process.

- Extensive experimental results on representative cross-modality medical image adaptation and segmentation tasks show that our UUDS outperforms state-of-the-art methods, demonstrating the advancements of our UUDS in addressing domain shift in a breakthrough non-adversarial manner.

## 2 RELATED WORK

**Unsupervised domain adaptation:** Unsupervised Domain Adaptation (UDA) plays an important role in medical image analysis, offering a promising approach to address domain shift challenges in medical image segmentation without necessitating labeled target data. Existing UDA methods (Yao et al., 2022) try to bridge the shift between source and target domains by aligning image distribution through adversarial learning. For instance, CycleGAN (Zhu et al., 2017) and CUT (Park et al., 2020) transfers the source domain to the target domain by harmonizing image appearance. DDC (Tzeng et al., 2014) prioritizes aligning feature distribution between the source and target domains. Dou et al. (Dou et al., 2019) proposed a method that aims to align feature spaces by using multiple scale feature information. Additionally, UDA methods such as CycADA (Hoffman et al., 2018) and SIFA (Chen et al., 2020) tackle domain shifts by addressing both image and feature distribution discrepancies. Bui et al. (Bui et al., 2020) introduced an effective method for image-to-image translation based on flow-based methods and deformation information. Dar et al. (Dar et al., 2019) proposed a novel approach for image synthesis in multi-contrast MRI based on generative adversarial network (GAN) architectures. Yurt et al. (Yurt et al., 2021) proposed a multi-stream generative adversarial network (mustGAN), for enhancing image synthesis in multi-contrast MRI via a mixture of multiple one-to-one streams and a joint many-to-one stream. Zhang et al. (Zhang et al., 2022) proposed a switchable CycleGAN model for image synthesis between multi-contrast brain MRI images, which outperforms the original CycleGAN on cross-contrast MRI image synthesis. Zou et al. (Zou et al., 2020) proposed a Dual-Scheme Fusion Network (DSFN) for unsupervised domain adaptation. DSFN builds both source-to-target and target-to-source connections to help reduce the domain gap to improve the network performance further. DAR-Net (Yao et al., 2022) integrated a 2D style transfer network with a 3D segmentation network to address the complexities of 3D medical images.

**Vision-Language Mode:** Vision-language models (VLMs) (Radford et al., 2021; He et al., 2020; Devlin, 2018) have made significant progress across various domains. Specifically, VLMs capture the correlation between vision and language through various cross-modal objectives, such as image-text contrastive learning, masked cross-modal modeling, image-to-text generation, and image-text/region-word matching. Early VLMs (Jia et al., 2021) typically employed a single pre-training objective. For example, different single-modal objectives have been explored to fully utilize the potential of each modality. For the image modality, this includes masked image modeling, while for the text modality, masked language modeling is employed. More recent VLMs (Yao et al., 2021) introduce multiple objectives (e.g., contrastive, alignment, and generative objectives) to leverage their synergy, resulting in more robust models and improved performance on downstream tasks. Yet, adapting VLMs to the medical domain presents significant challenges, largely due to domain-specific obstacles such as the use of proprietary datasets, the need for fine-grained medical knowledge, and the inherent difficulty in generalizing across diverse medical domains and tasks.

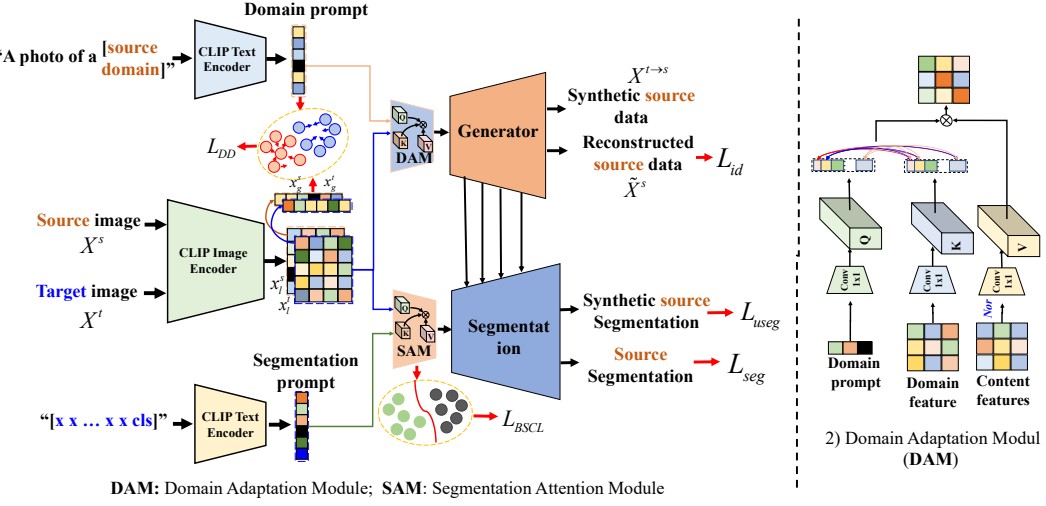

DAM: Domain Adaptation Module; SAM: Segmentation Attention Module

1) Unified Uncertain Dual-prompts cross-domain Segmentation framework (UUDS)

Figure 1: 1) UUDS creates a unified framework where segmentation and domain adaptation are seamlessly combined for unsupervised segmentation-aware cross-domain medical image segmentation through dual prompts. 2) DAM creatively utilizes the domain prompt to guide domain adaptation, enabling non-adversarial domain adaptation.

## 3 METHOD

Our Unified Uncertain Dual-prompts cross-domain Segmentation framework, named UUDS, forms a unified segmentation-aware unsupervised cross-domain segmentation framework by integrating domain adaptation and segmentation models (Figure. 1). Advanced than existing methods, UUDS enables the model to leverage semantic information from segmentation feature representation space to guide domain adaptation, ensuring that the model preserves sensitivity to segmentation tasks throughout the whole adaptation process. Formally, given the source imaging (**Source domain**) with corresponding interesting object label $\{X^s, Y^s\} \in R^{C \times H \times W}$, and unlabeled target imaging $X^t \in R^{C \times H \times W}$ (**Target domain**). The goal is to segment the same object from $X^t$. To this end, UUDS transfer $x^t$ to synthetic source imaging $x^{t \rightarrow s}$ for overcoming the domain shift between $x^s$ and $x^t$. Meanwhile, the segmentation module predicates the segmentation results of $x^{t \rightarrow s}$ by using the features of synthetic source imaging $x^{t \rightarrow s}$ from generator.

### 3.1 DUAL-PROMPTS LEANING

To achieve the domain adaptation from target domain to source domain, UUDS leverages text embeddings encoded by CLIP to bridge the cross-modality embedding spaces. Yet, as a VLM trained on image-level alignment tasks, CLIP-based models are good at capturing the global-level domain style features while showing insufficient capability to fully capture the region-level content of medical images due to the complexity of anatomical structures and the intricate details inherent in medical imaging. To this end, our UUDS proposes a dual-prompt system to capture global and region-level information in a disentangled manner by leveraging both a hard prompt and a soft prompt. The domain prompt is a hard prompt encoded by CLIP text encoder, focusing on learning domain-invariant features for domain-adaptation tasks, covered in more detail next in subsection 3.1.1. And the soft prompt is randomly initialized like CoOP (Zhou et al., 2022), using a few tunable continuous vectors to capture region level or lesion features vital to segmentation tasks. The reason that we choose soft prompts rather than hard prompts is due to the shape and characteristics of lesions and tumors being hard to describe in natural language. In medical image analysis, many lesions and abnormalities are uneven and vary in shape. Therefore, applying continuous and tunable soft prompts are better for this situation.

After all, UUDS employs dual prompts, domain and segmentation prompts, to facilitate cross-domain invariant information learning. The domain prompt captures domain-specific information, while the segmentation prompt focuses on learning cross-domain invariant segmentation features.

### 3.1.1 DOMAIN PROMPT FOR DOMAIN ADAPTATION

Formally, as shown in Figure. 1, given $X^s, X^t$, the CLIP text encoder converts the domain prompt, 'A photo of a [source domain]', into text embedding $T^d$, which describes the domain distributions of source data from a global aspect. At the same time, a learnable segmentation prompt $T^s$ is initialized to learn the cross-domain invariant segmentation information. The image encoder extracts the content and domain-specific feature representation $\{X_c^s, X_d^s\} \in R^{c \times h \times w}, \{X_c^t, X_d^t\} \in R^{c \times h \times w}$ from $X^s, X^t$, respectively. The content feature representation $X_c^i$ is extracted from the last layer of CLIP image encoder, the domain feature representation $X_d^i$ is learned from ViT layer of CLIP image encoder. To ensure that the domain prompt effectively learns the domain distribution. The global information $\{x_g^s, x_g^t\} \in \mathbb{R}^c$ are sampled from $\{X_d^s, X_d^t\}$ are used for domain distillation.

To align the domain prompts with source and target images, respectively, We use contrastive learning by maximizing the *cosine* similarity between $x_g^s$ and $T^d$ and minimizing the the *cosine* similarity between $x_g^t$ and $T^d$ through $L_{DD}$.

$$L_{DD}(x_g^s, x_g^t, T^d) = \log\left(\frac{exp((1 - sim(x_g^s, T^d))/\tau}{exp(1 - sim(x_g^s, T^d))/\tau + exp(1 - sim(x_g^t, T^d))/\tau}\right) \quad (1)$$

where $sim$ represents feature cosine similarity measurement, $\tau$ is a temperature hyper-parameter. $L_{DD}$ distills domain information from the cross-modality space, ensuring that $T^d$ learns the source domain distribution and maintains consistency within the intra-domain distribution. What's more, the domain consistency in synthetic imaging $X^{t \to s}$ is also ensured.

$$L_{DD}(x_g^{t \to s}, x_g^t, T^d) = \log\left(\frac{exp(1 - sim(x_g^{t \to s}, T^d))/\tau}{exp(1 - sim(x_g^{t \to s}, T^d))/\tau + exp(1 - sim(x_g^t, T^d))/\tau}\right) \quad (2)$$

More specifically, the domain information from source and target imaging is first fused with the image features by the Domain Adaption Module(DAM) that will fuse features in the early stage. This early fusion will increase the alignment of finer-grained features, as suggested in GLIP (Li* et al., 2022) and Grounding Dino(Liu et al., 2023). Based on the domain prompt $T^d$, domain feature representation $X_d^i$, and content information $X_c^i$, Domain Adaptation Module (DAM) reconstructs the domain distribution of each content feature representation. DAM first utilizes three convolution layers to project $T^d$ into sequence $Q \in \mathbb{R}^c$ project $X_d^i$ into sequences $K \in \mathbb{R}^{c \times h \times w}$, and project $X_c^i$ into sequences $V \in \mathbb{R}^{c \times h \times w}$, where $Q$ is the input Query sequence, $K$ and $V$ are the input Key,Value sequence. The cross-domain attention performs domain adaptation based on $Q$, $K$, and $V$.

$$x^d = \text{softmax}\left(\frac{QK^T}{d}\right)Nor(V) \quad (3)$$

where $d$ is a learnable scaling parameter to control the magnitude of the dot product. $Nor$ represents the normalization operation, which uses mean and variance to eliminate original domain information. Afterward, the adapted content $x^d$ is fed into the decoder to generate synthetic source data $X^{t \to s}$, meanwhile,reconstruct the source imaging $\tilde{X}^s$.

### 3.1.2 SEGMENTATION PROMPT FOR SEMANTIC-AWARE SEGMENTATION

The segmentation prompt $T^s$ captures the semantics of the segmented object, enabling the extraction of cross-domain invariant feature representations. Since the segmentation object is difficult to define explicitly for specific tasks, we use a trainable prompt $T^s$ to learn such anatomical characteristics. We follow the setting in (Zhou et al., 2022) to initialize our segmentation prompts as a combination of continuous vectors and class names. Instead of using "a photo of a" as the context, we introduce $M$ learnable context vectors, $\{v_1, v_2, \ldots, v_M\}$, each having the same dimension with the word embeddings in Clip Text encoder. For our tasks, the class names are typically segmentation target.

Formally, as shown in Figure. 1, given $T^s$, and the content feature representation $X_c^s, X_c^t$, $T^s$ learns the semantic information of the segmented object through in-batch contrastive learning. Specifically,

the segmentation prompt is first fused with the imaging content through the segmentation attention module (SAM) as described below. SAM works quite similarly to the DAM module above. First utilizes three convolution layers to project $T^d$ into sequence $Q \in \mathbb{R}^c$ project $X_c^i$ into sequences $K \in \mathbb{R}^{c \times h \times w}$ and $V \in \mathbb{R}^{c \times h \times w}$.

To establish the relationship between the segmentation prompt and the segmentation feature representation, in-batch contrastive learning is employed. Specifically, the updated content feature representation is projected into semantic feature embeddings $\{z_1, z_2, z_3, ..\}$ using multi-layer perceptrons (MLPs). These embeddings $\{z_1, z_2, z_3, ..\}$ are then classified into two categories based on the label: those containing the segmentation object and those not containing the segmentation object. This process is achieved by in-batch supervised contrastive loss $L_{BSCL}$.

$$L_{BSCL} = \sum_{i \in B} \frac{-1}{|P(i)|} \sum_{p \in P(i)} \log \frac{\exp\left(z_i \cdot z_p / \tau\right)}{\sum_{a \in A(i)} \exp\left(z_i \cdot z_a / \tau\right)} \tag{4}$$

Here, $B$ represents the sample in the input, $P(i)$, $A(i)$ is the set of indices of all images with /without segmentation object in the input. The $\cdot$ symbol denotes the inner (dot) product, $\tau$ is a scalar temperature parameter.

Afterward, by combining the multi-level features learned from the generator, these features are then input into the segmentation decoder for segmentation. This novel and unified framework situates domain adaptation within the segmentation space, ensuring a focus on segmentation-aware features during the domain adaptation, and enhancing the sensitivity of domain adaptation to segmentation-specific features.

## 3.2 UNCERTAINTY ESTIMATION

To address the limitation of existing unsupervised domain adaptation methods are unable to focus on the specific features crucial for segmentation tasks while labeling is unavailable. The domain adaptation between source and target images is directly evaluated by approximating the uncertainty of segmentation on synthetic source imaging $X^{t \to s}$. Since no label is available for $X^{t \to s}$, the uncertainty of each pixel $\hat{Y}_{i,j}^{t \to s} \in X^{t \to s}$ is computed through predictive entropy, where predictions with high entropy indicate uncertainty in segmentation map.

$$u_{i,j} = -\hat{Y}_{i,j}^{t \to s} \log(\hat{Y}_{i,j}^{t \to s} + \varepsilon) \tag{5}$$

Based on the uncertainty value, the pseudo label $\hat{Y}^{t \to s}$ of $X^{t \to s}$ can be obtained by removing those uncertainty predictions.

$$Y_{i,j}^{t \to s} = \{\hat{Y}_{i,j}^{t \to s} | \mu_{i,j} < \chi\} \tag{6}$$

where $\varepsilon = 1e^{-9}$ to avoid singularity, $\chi$ is a threshold for selecting the uncertain labels.

Yet, the pseudo label $Y^{t \to s}$ cannot fully and confidently represent the segmentation performance at each pixel. To maximize the effectiveness of the pseudo label, partial information from $Y^{t \to s}$ is used to supervise the domain adaptation process partially. Specifically, based on the uncertainty value of prediction, the certain region $Y^{mask}$ of prediction can be computed and constructs a partial label.

$$Y_{i,j}^{mask} = \begin{cases} 1, & \text{if } \mu_{i,j} < \chi \\ 0, & \text{otherwise} \end{cases} \tag{7}$$

Based on $Y^{t \to s}$ and $Y^{mask}$, the segmentation performance on $X^{t \to s}$ is evaluated by segmentation loss $L_{seg}$ to guide the domain adaptation.

$$L_{useg}(\hat{Y}^{t \to s}, Y^{t \to s}) = 1 - \frac{2 \sum\limits_{i,j \in Y^{mask}} Y_{i,j}^{t \to s} \times \hat{Y}_{i,j}^{t \to s}}{\sum\limits_{i,j \in Y^{mask}} Y_{i,j}^{t \to s} + \sum\limits_{i,j \in Y^{mask}} \hat{Y}_{i,j}^{t \to s}} \tag{8}$$

Additionally, the segmentation performance on true $X^s$ is also evaluated by segmentation loss $L_{seg}(\hat{Y}^s, Y^s)$ to ensure the segmentation semantic consistency in true source imaging.

## 3.3 MODEL TRAINING

In summary, during training, a total of four types of losses are used to supervise our UUDS.

$$L_{total} = \alpha \left( L_{DD}(x_g^s, x_g^t, T^d) + L_{DD}(x_g^{t \rightarrow s}, x_g^t, T^d) \right) + \beta L_{BSCL}(X^s, Y^s)$$
$$+ \gamma \left( L_{useg}(\hat{Y}^{t \rightarrow s}, Y^{t \rightarrow s}) + L_{seg}(\hat{Y}^s, Y^s) \right) + \lambda L_{id}(\tilde{X}^s, X^s)$$

where $\alpha$, $\beta$, $\gamma$, and $\lambda$ represent the weight coefficients. $L_{id}$ is identical loss, which computes the difference between reconstructed $\tilde{X}^s$ and $X^s$ at the pixel level. The definition of $L_{id}$ is:

$$L_{id}(\tilde{X}^s, X^s) = \left\| \tilde{X}^s - X^s \right\|_1 \tag{9}$$

Based on the above loss function, both domain adaptation and segmentation models are joint trained in an end-to-end manner.

## 4 EXPERIMENTS

### 4.1 DATASET AND IMPLEMENTATION DETAILS

**1) BraTS Dataset:** The multi-modal Brain Tumor Segmentation (BraTS) challenge 2020 dataset (Menze et al., 2014) includes spatially aligned MRI scans from 369 patients, covering four modalities (T1, ceT1, T2, and FLAIR) with a resolution of 1.0 $mm^3$ and an in-plane size of 240 × 240 pixels. Since the ground truth for the official validation and testing sets is not publicly available, we conducted our experiments using the official training set. Following previous works (Wu et al., 2024), in our unsupervised cross-modality segmentation task, we also focused on segmenting the whole tumor using T2 and FLAIR images. We treated images from one modality of 143 patients as the source domain and images from the other modality of another 143 patients as the target domain, in each direction. We used 42 images (21 for each modality) for validation and 41 images in the target domain for testing. In the preprocessing step, we truncate the pixel value by the 5%, 95% percentage of min-max value and normalized the intensity of each modality to [-1, 1].

**2) Vestibular Schwannoma Segmentation Dataset:** Vestibular Schwannoma (VS) segmentation dataset (Shapey et al., 2021) includes 3D MRI images from 242 patients. Each patient was scanned using contrast-enhanced T1-weighted (ceT1) and high-resolution T2-weighted (hrT2) MRI, with an in-plane resolution of approximately 0.4 mm × 0.4 mm, an in-plane size of 512 × 512, and a slice thickness of 1.5 mm. These two modalities were used for bidirectional adaptation, where ceT1 and hrT2 served as the source and target domains, respectively. Following the setup of the Cross-modality Domain Adaptation Challenge 2021 (Shapey et al., 2021), the validation set from the target domain was used to tune hyperparameters, and the testing set was reserved solely for the final inference. For data preprocessing, each image was normalized based on its intensity mean and standard deviation. Following previous works (Wu et al., 2024), the dataset was randomly split into 200 patients for training, 14 patients for validation, and 28 patients for testing. For the training set, images from one modality of 100 patients were used as the source domain, while images from the other modality of the remaining 100 patients were used as the target domain.

**3) Implementation Details:** The ResNet version of CLIP is chosen as the backbone, and all parameters, including CLIP text encoder and image encoder, are fine-tuned during training. The source and target decoders have the same structure and are constructed by Resblock and upsample layers, which gradually upsample the features until they are the same size as the input image. The framework is implemented on PyTorch and utilizes four L40s GPUs with 46 GB of memory each. The Adam optimizer was used for optimization, with the learning rate set to 1e-4 and weight decay to 0.01. The weight coefficients are set as $\alpha = 1.0$, $\beta = 1.0$, $\gamma = 1.0$, $\lambda = 1.0$. Remarkably, all datasets' domain prompt input is set as "A photo of a source domain" while learning.

### 4.2 COMPARISON WITH STATE-OF-THE-ART UDA METHODS

To assess the effectiveness of our UUDS, we compared it with state-of-the-art UDA methods, including ADVENT (Vu et al., 2019), SIFA (Chen et al., 2020), CUT (Park et al., 2020), AccSeg (Zhou

Table 1: Quantitative comparison of various UDA methods on glioma segmentation.

| Method | FLAIR → T2 | | T2 → FLAIR | |
|---|---|---|---|---|
| | Dice (%) | ASSD (mm) | Dice (%) | ASSD (mm) |
| w/o DA (**Lower bound**) | 47.16±24.39 | 20.82±11.31 | 68.46±21.74 | 8.71±8.38 |
| Labeled target (**Upper bound**) | 81.18±16.62 | 3.95±8.28 | 84.50±15.41 | 3.73±6.48 |
| ADVENT (Vu et al., 2019) | 39.83±24.07 | 16.76±8.43 | 55.03±23.34 | 10.51±8.79 |
| SIFA (Chen et al., 2020) | 55.52±20.30 | 14.77±9.06 | 66.03±14.34 | 7.45±4.38 |
| CUT (Park et al., 2020) | 66.03±25.81 | 9.79±13.95 | 72.33±21.94 | 7.21±12.43 |
| AccSeg (Zhou et al., 2021) | 63.95±15.93 | 17.52±8.69 | 69.81±22.06 | 8.98±6.91 |
| HRDA (Hoyer et al., 2022) | 27.48±18.39 | 27.52±10.31 | 63.06±14.65 | 13.63±6.37 |
| CDAC (Wang et al., 2023) | 25.55±14.11 | 33.61±10.24 | 21.40±9.83 | 38.96±7.88 |
| UUDS (Our) | **69.83±13.40** | **5.51±3.26** | **75.22±12.12** | **5.99±2.48** |

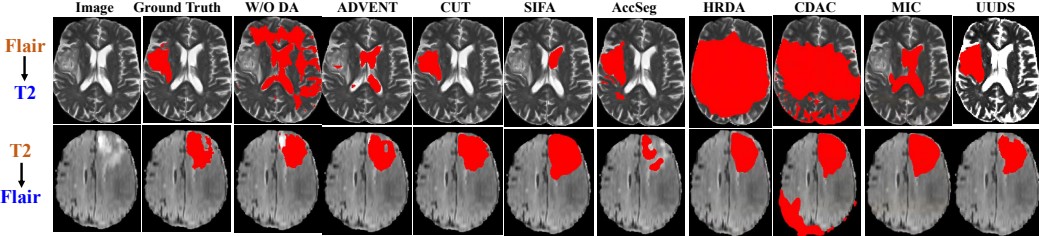

Figure 2: Visualization of segmentation results obtained by different UDA methods on the BraTS dataset.

et al., 2021), HRDA (Hoyer et al., 2022), CDAC (Wang et al., 2023), MIC (Hoyer et al., 2023). Additionally, following prior works, we evaluated the impact of domain shift by applying the additional segmentation model trained on source data directly to the target domain for segmentation. This performance, without domain adaptation ("w/o DA"), represents the **Lower Bound**. Conversely, the performance of segmentation model trained with labeled target domain data is considered the **Upper Bound**.

### 4.2.1 PERFORMANCE ON BRATS DATASET

Table 1 presents the quantitative results of state-of-the-art UDA methods on the BraTS dataset. The substantial performance gap between the lower and upper bounds underscores the significant domain shift between the T2 and FLAIR modalities, which has a major impact on tumor segmentation. In comparison to other state-of-the-art UDA methods, our UUDS achieved the best performance, with Dice scores of 69.83% and 75.22% and ASSD scores of 5.51mm and 5.99mm on the T2 and FLAIR modalities, respectively. This further highlights the effectiveness of UUDS in unsupervised cross-modality segmentation tasks. These high performances are attributed to the effectiveness of our unified framework, which directly utilizes segmentation information to supervise the domain adaptation. Figure.2 shows the segmentation performance on tumor segmentation. We can notice that our method achieves more accurate and smoother tumor segmentation compared to state-of-the-art UDA methods on both T2 and FLAIR images. It is evident that existing methods struggle to segment the entire tumor across different modalities. This suggests that the feature distribution of tumors varies significantly between modalities, and current UDA methods lack sensitivity to local feature representation. In contrast, our model outperforms these state-of-the-art UDA approaches, demonstrating higher sensitivity to domain distribution, even in small tumor regions. Moreover, the smoother and more precise tumor boundaries produced by our model further validate its effectiveness in overcoming domain shift, particularly in local tumor.

### 4.2.2 PERFORMANCE ON VESTIBULAR SCHWANNOMA (VS) SEGMENTATION

Two unsupervised cross-modality segmentation tasks on the VS dataset are used to evaluate our UUDS, 1) ceT1 to hrT2 and 2) hrT2 to ceT1. Table 2 shows the segmentation performance from

Table 2: Quantitative comparison of various UDA methods on VS segmentation.

| Method | ceT1 → hrT2 | | hrT2 → ceT1 | |
|---|---|---|---|---|
| | Dice (%) | ASSD (mm) | Dice (%) | ASSD (mm) |
| w/o DA (Lower bound) | 0.00±0.00 | 48.30±5.29 | 2.65±8.18 | 31.01±16.61 |
| Labeled target (Upper bound) | 88.17±7.81 | 1.03±2.67 | 90.72±12.47 | 0.30±0.53 |
| ADVENT (Vu et al., 2019) | 5.36±9.61 | 35.68±11.49 | 21.94±23.07 | 34.11±15.24 |
| CUT (Park et al., 2020) | 73.64±15.57 | 3.96±6.86 | 56.27±31.37 | 9.25±17.14 |
| SIFA (Chen et al., 2020) | 69.75±21.54 | 6.01±5.88 | 67.48±20.32 | 6.51±8.89 |
| AccSeg (Zhou et al., 2021) | 30.95±31.81 | 15.44±10.63 | 37.01±31.97 | 17.06±21.11 |
| HRDA (Hoyer et al., 2022) | 6.15±13.38 | 21.69±16.67 | 17.72±19.74 | 14.69±11.48 |
| CDAC (Wang et al., 2023) | 0.32±1.38 | 25.39±11.00 | 2.98±8.13 | 35.54±18.57 |
| MIC (Hoyer et al., 2023) | 54.82±24.55 | 11.84±11.66 | 13.44±22.95 | 30.13±22.37 |
| UUDS (Our) | 67.95±14.92 | 4.64±3.21 | **68.87±19.43** | **4.30±7.15** |

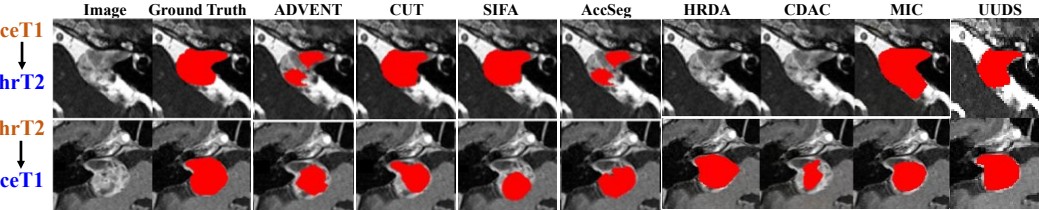

Figure 3: Visualization of segmentation results obtained by different UDA methods on the VS segmentation.

state-of-the-art UDA methods. The segmentation model trained on hrT2 achieved strong results (Dice 88.17%) when segmenting tumors from true hrT2 images. However, it struggled significantly with detecting and segmenting VS from ceT1 images, resulting in a Dice score of 0.00%. The model trained on hrT2 showed similar difficulties with ceT1, underscoring the significant impact of domain shift between ceT1 and hrT2 on performance. In contrast, our UUDS outperforms existing methods, achieving superior Dice scores of 68.87% and ASSD scores of 4.30mm for ceT1 segmentation. This highlights UUDS's effectiveness in combining the segmentation and domain adaptation for overcoming domain shift, also demonstrating its advanced ability to learn domain distributions and disentangle domain information from content. Figure 3 presents representative segmentation results by various state-of-the-art UDA methods on the VS dataset. It is evident that our UUDS produced more accurate segmentation outcomes for both ceT1 and hrT2 images. Notably, existing adversarial learning-based methods, such as CDAC (Wang et al., 2023), performed poorly on segmentation, highlighting the limitations of existing methods in learning domain-discriminative features and handling challenges in learning domain specific information. Our UUDS demonstrates strong alignment and consistency with ground truth. This highlights the advancements of UUDS in tackling domain adaptation challenges and its superior ability to learn domain-specific representations. Moreover, the higher performance shows the advancement of our model in learning segmentation-aware feature representation.

### 4.3 ANALYSIS OF UUDS

To further evaluate the effectiveness of each module in our design, we conducted ablation experiments as shown in Table 3. We performed experiments to assess the impact of the domain prompt, segmentation prompt, and Uncertainty estimation individually. The results demonstrate that each module plays a significant role in the overall performance. Notably, the model's performance deteriorates significantly when any of the prompts is omitted. We will elaborate the experiments in detail for each component in following sub-sections.

**1) Domain prompt effective in domain adaptation:** The effectiveness of domain prompt is further assessed using the paired BraTS (FLAIR, T2) dataset. As shown in Figure 4, a significant domain shift is observed between the FLAIR and T2 modalities. After applying the domain promptly, the

Table 3: Ablation study results: each row shows the performance of different combinations of components. A checkmark indicates that the component is present, while a cross means the component is ablated.

| Dual prompts | | Uncertainty estimation | T2→Flair | |
| Domain | Segmentation | | Dice (%) | ASSD (mm) |
| --- | --- | --- | --- | --- |
| ✓ | ✓ | ✓ | **75.22±12.12** | **5.99±2.48** |
| ✓ | ✓ | × | 69.36±15.67 | 5.66±4.01 |
| ✓ | × | ✓ | 64.03±23.08 | 6.49±5.95 |
| × | ✓ | ✓ | 63.91±19.46 | 4.10±2.55 |

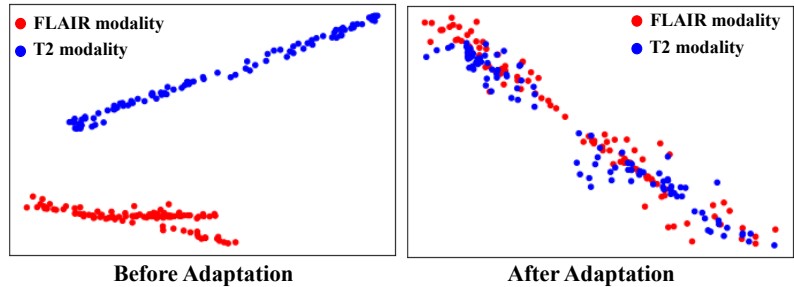

Figure 4: The t-SNE visualization illustrates the distribution of the FLAIR and T2 datasets before and after domain adaptation by domain prompt.

target domain is effectively adapted to the source domain, reducing the domain shift and demonstrating the model's ability to learn and adapt to domain distributions; Therefore, although the domain prompt does not directly participate in the segmentation model, it still plays a crucial role by capturing the feature shifts from the source to the target domain. Additionally, in table 3, we observed a significant drop in Dice results once the Domain prompts were removed. This observation articulates the fact that the domain prompts play a vital part in our framework.

**2) Segmentation prompt sensitives to semantic representation:** We also analyzed the effect of the segmentation prompt. As mentioned above, the segmentation prompt is dedicated to capturing regional features, such as lesions, tissues, and other anatomical characteristics. Therefore, we expect the segmentation prompts can sharply increase the Dice results for segmentation. From table 3, we can see that the segmentation results drop more than 10% in terms of Dice, from 75.22% to 64.03%. These experiments clearly exhibit the importance of the segmentation prompts.

**3) Uncertainty estimation for segmentation optimization:** As discussed in previous sections, uncertainty estimation helps overcome the limitation of using unlabeled target data for direct supervision in the cross-domain adaptation process. Our ablation experiment further demonstrates that uncertainty estimation improves segmentation performance. As shown in Table 3, removing the uncertainty estimation module resulted in a significant drop in segmentation performance, with the Dice score decreasing by more than 5%, from 75.22% to 69.36%.

## 5 CONCLUSION

As the first end-to-end framework that unifies segmentation and domain adaptation, our experiments validate the hypothesis that feedback from the segmentation model is essential in the domain adaptation process. We innovatively leverage the cross-domain invariance of vision-language models (VLMs) to bridge the gap between the two domains and employ a dual prompts system to simultaneously learn domain-invariant style and content features. Extensive experiments demonstrate the effectiveness of our dual-prompts method. To address the challenge of missing labels in the target domain, we introduce uncertainty estimation, which further enhances the stability of our segmentation results. We achieved state-of-the-art performance on multiple public datasets, and ablation studies confirm the importance of each module. We hope our work will inspire future research to recognize that domain adaptation and segmentation can be unified within a single framework.

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
