# OpenReview forum: "Unified Uncertain Dual-prompts cross-domain Segmentation framework for medical image segmentation"
_ICLR.cc/2025/Conference — ICLR 2025 Conference Withdrawn Submission_

### Official Review · Reviewer_xCfd · 2024-10-31

**Soundness:** 3
**Presentation:** 4
**Contribution:** 3
**Rating:** 3
**Confidence:** 5

**Summary:**

This paper proposed an unsupervised domain adaptation model for medical image segmentation. The key designs are to leverage text prompts including domain and task texts and combine both image transformation and segmentation together. The proposed model achieves good performance on two public datasets. Overall, the designs are reasonable and the experiments demonstrate the effectiveness of the proposed designs.

**Strengths:**

1. Good performance for the UDA segmentation tasks on two datasets.
2. Reasonable designs. Most modules are supported by corresponding experiments.

**Weaknesses:**

Model Claims and Uniqueness:
While the paper claims to offer a unified model that integrates domain adaptation and image segmentation as a single approach rather than as “two separate parts,” it remains unclear how it is achieved. What sets the proposed model different from previous methods? The uniqueness of this approach may be over-claimed, as the use of two types of text prompts or sharing features does not seem sufficient to innovatively combine the two tasks together.

Figure 1 Clarity:
Figure 1 lacks clarity on several points. Are the CLIP encoders kept frozen? What specific architecture is used for the SAM design? Additionally, the interaction between the generation and segmentation decoders is not clearly depicted. Providing more details on these parts can further improve understanding.

Baseline Performance for Glioma Segmentation:
In glioma segmentation, the baseline performance (Lower Bound) has surpassed many comparable methods. Why? Including more recent or comparable works would be valuable to support the proposed model.

Use of Text Prompts and Clinical Relevance:
The use of text prompts is not novel in medical image segmentation, raising questions about whether CLIP can accurately capture domain differences for medical data. Exploring alternative options, such as medical-specific text prompts or just one-hot codes, might yield further insights. Additionally, from a clinical perspective, what is the proposed domain-invariant feature? How to support this claim? Why does it work well for UDA tasks? If there is no unique clinical design, more datasets like natural images should be included.

**Questions:**

See the above weaknesses.
Overall, please
1. Improve the statements of contributions and existing challenges. Do not over-claim.
2. Clarify the baseline performance.
3. Improve Fig. 1.
4. Discuss the usage of CLIP prompts and other prompts.
5. How is the performance of image synthesis? Is it useful to include image segmentation for MRI sequence transformation?
6. Compared to other foundation models, what is the superiority? We can see that SynthSeg can achieve cross-modal segmentation and the SAM model also has been widely used in medical image segmentation.

---

### Official Review · Reviewer_CvJ2 · 2024-11-03

**Soundness:** 1
**Presentation:** 2
**Contribution:** 1
**Rating:** 1
**Confidence:** 5

**Summary:**

This paper proposed a method named Unified Uncertain Dual-prompts cross-domain Segmentation (UUDS) for unsupervised domain adaptation of (cross-modality) medical image segmentation. UUDS uses CLIP to leverage domain prompt and segmentation prompt to extract domain invariant representation. UUDS also applies the uncertainty-regularized pseudo-labeling method.

**Strengths:**

1. This submission targets an important problem, i.n., cross-modality medical image domain adaptive segmentation.

**Weaknesses:**

The reviewer found many flaws in the submission, specifically:
1. Section 2, Related Works Unsupervised domain adaptation (page 3, lines 129-149), is significantly biased and not comprehensive enough. The related UDA works merely focused on GAN-based methods while completely ignoring many significant branches. For instance, semi-supervised learning with pseudo-labeling has represented many advances in recent years, but it was not introduced in this section. In fact, this manuscript utilized pseudo-labeling (Sec. 3.2, Eq. 8) while not acknowledging this kind of method in the related works.
2. The reviewer has reason to believe that this paper **selectively** presented previously reported results from existing literature, which, to the reviewer's mind,  is a **serious** problem that could directly lead to a rejection in a conference like ICLR. Please find my detailed proof below:
* This paper used the same datasets as the reference FPL+ [1].
* Table 1 in this submission corresponds to Table 2 in [1]. It is clear that the results of w/o DA, Labeled target, ADVENT, SIFA, CUT, AccSeg, HRDA, and CDAC are identical to the results reported in [1]. However, the current submission **removed** the results from methods that are **better than the proposed UUDS**, including CycleGAN, DAR-NET, and FPL (please refer to details in Table 2 of [1]).
* Table 2 in this submission corresponds to Table 1 in [1]. It is clear that the results of w/o DA, Labeled target, ADVENT, SIFA, CUT, AccSeg, HRDA, CDAC, and MIC are identical to the results reported in [1]. However, the current submission **removed** the results from methods that are **better than the proposed UUDS**, including CycleGAN, DAR-NET, and FPL (please refer to details in Table 1 of [1]).
* [1]: Wu, Jianghao, et al. "FPL+: Filtered Pseudo Label-based Unsupervised Cross-Modality Adaptation for 3D Medical Image Segmentation." IEEE Transactions on Medical Imaging (2024).
3. Even without considering the fact that this submission **selectively/manipulatively** presents existing results, the results of UUDS are fairly close to methods published in 2020 (i.e., CUT/SIFA) in Table 1/2, which indicates the proposed method does not bring considerable improvement.
4. This paper seems to overlap with a CVPR 2024 paper [2] in terms of using CLIP and prompts for UDA. Although [2] targets classification, the similar idea of using CLIP and dual/mutual prompts shall be discussed.
* [2]: Du, Zhekai, et al. "Domain-agnostic mutual prompting for unsupervised domain adaptation." Proceedings of the IEEE/CVF Conference on Computer Vision and Pattern Recognition. 2024.
5. The presentation is not clear and is hard to follow, especially in Section 3.
6. Many contributions are not justified/vague:
* Contribution 1: how is the 'unified' defined in this paper? In many pseudo-labeling-based methods, UDA and segmentation are combined/unified. So it is not a contribution from UUDS.
* Contribution 4: uncertainty-regularized pseudo-labeling is a mature technique in semi-supervised learning; how could it be a contribution of this paper?
* Contribution 5: refer to my previous comments; the proposed UUDS did not set a new SOTA at all.

Minor errors/typos: This is not an exhaustive list, as the reviewer is not a proofreader. However, extensive minor errors make me worry about whether the manuscript is carefully polished.
1. Page 3, line 108, 'the largely vision-language mode' is weird.
2. Page 3, line 117, 'novel using uncertainty estimation to' reads weird.
3. Page 4, Line 201, '3.1 Dual-prompts leaning' -> should be learning.
4. Page 5, Line 267, 'are typically segmentation target', reads weird.
5. In Table 1 and Figure 2, the presenting methods are not corresponding, i.e., there are results of MIC in Figure 2, while the results of MIC are not included in Table 1.

**Questions:**

About domain prompts, why only use 'A photo of a source domain', as stated on Page 7, line 373?

Why don't also input the prompt 'A photo of a target domain', and then maximize the cosine similarity between $x_g^t$ and $T^d$, and minimize the cosine similarity between $x_g^s$ and $T^d$ (i.e., another form of Eq. (1) $L_{DD}$)?

**Details Of Ethics Concerns:**

This submission **selectively/manipulatively** presents existing results (i.e., it cited results from existing literature. However, it removed the results that are better than the proposed method).

The reviewer understands this problem is not as serious as plagiarism, but I believe an ICLR paper should not do that. Please find details/proof below (duplicated from my Weakness section).

The reviewer has reason to believe that this paper **selectively** presented previously reported results from existing literature, which, to the reviewer's mind,  is a **serious** problem that could directly lead to a rejection in a conference like ICLR. Please find my detailed proof below:
* This paper used the same datasets as the reference FPL+ [1].
* Table 1 in this submission corresponds to Table 2 in [1]. It is clear that the results of w/o DA, Labeled target, ADVENT, SIFA, CUT, AccSeg, HRDA, and CDAC are identical to the results reported in [1]. However, the current submission **removed** the results from methods that are **better than the proposed UUDS**, including CycleGAN, DAR-NET, and FPL (please refer to details in Table 2 of [1]).
* Table 2 in this submission corresponds to Table 1 in [1]. It is clear that the results of w/o DA, Labeled target, ADVENT, SIFA, CUT, AccSeg, HRDA, CDAC, and MIC are identical to the results reported in [1]. However, the current submission **removed** the results from methods that are **better than the proposed UUDS**, including CycleGAN, DAR-NET, and FPL (please refer to details in Table 1 of [1]).
* [1]: Wu, Jianghao, et al. "FPL+: Filtered Pseudo Label-based Unsupervised Cross-Modality Adaptation for 3D Medical Image Segmentation." IEEE Transactions on Medical Imaging (2024).

---

### Official Review · Reviewer_qCE8 · 2024-11-03

**Soundness:** 3
**Presentation:** 3
**Contribution:** 3
**Rating:** 3
**Confidence:** 3

**Summary:**

The authors propose a unified framework for unsupervised domain adaptation for medical image segmentation. The method is an end-to-end framework by integrating domain adaptation and segmentation models. The authors utilize the cross-domain invariance of vision-language models (VLMs) to bridge the gap between the two domains, implementing a dual prompt system to concurrently learn domain-invariant style and content features.

**Strengths:**

1. The paper proposes an end-to-end framework for unsupervised domain adaption.
2. CLIP is employed to unsupervised cross-domain medical image segmentation and address the domain gap challenge on natural images to the medical image field.
3. The paper proposes Dual-prompts to learn domain and segmentation invariant representation learning.

**Weaknesses:**

1. The paper exclusively compares its results with UDA methods applied to natural images. Could the authors include recent UDA methods relevant to medical images, such as FPL-UDA [1] and FPL+ [2]?
2. In comparison to the first two tables presented in FPL+ [2], the paper omits results for CycleGAN, DAR-NET, FPL, and FPL+. These methods demonstrate superior performance relative to the proposed approach. Specifically, FPL+ achieves Dice scores of 75.76% and 84.81% for the FLAIR to T2 and T2 to FLAIR conversions, respectively, while the proposed method only attains Dice scores of 69.83% and 75.22%. This indicates a significant drop in performance for the proposed method. Could the authors clarify why these results were not included?
3. Lack of generalization. Can the authors provide additional datasets, such as the MMWHS Dataset?
4. FPL+ [2] proposes a one-stage joint learning method to train a final segmentator. Meanwhile, FPL+ is also based on uncertainty estimation. The two modules are similar to your proposed ideas.

[1] Wu, Jianghao, et al. "Fpl-uda: Filtered pseudo label-based unsupervised cross-modality adaptation for vestibular schwannoma segmentation." 2022 IEEE 19th International Symposium on Biomedical Imaging (ISBI). IEEE, 2022.

[2] Wu, Jianghao, et al. "FPL+: Filtered Pseudo Label-based Unsupervised Cross-Modality Adaptation for 3D Medical Image Segmentation." IEEE Transactions on Medical Imaging (2024).

**Questions:**

Please see the weakness.

---

### Note · Authors · 2024-11-12

I have read and agree with the venue's withdrawal policy on behalf of myself and my co-authors.